# Differences in the predicted nasoseptal flap length among races: A propensity score matching analysis

Chang Yeong Jeong[1], Jin Hee Cho[1], Yong Jin Park[1], Sung Won Kim[1], Jae-Sung Park[2], Mohammed Abdullah Basurrah[3], Do Hyun Kim[1]*, Soo Whan Kim[1]*

1 Department of Otolaryngology-Head and Neck Surgery, Seoul St. Mary's Hospital, College of Medicine, The Catholic University of Korea, Seoul, Korea, 2 Department of Neurosurgery, Seoul St. Mary's Hospital, College of Medicine, The Catholic University of Korea, Seoul, Korea, 3 Department of Surgery, College of Medicine, Taif University, Taif, Saudi Arabia

\* dohyuni9292@naver.com (DHK); kshent@catholic.ac.kr (SWK)

## Abstract

### Objectives

We compared the lengths of a nasoseptal flap (NSF) and skull base according to race, age, and sex.

### Methods

We performed paranasal sinus computed tomography in 19,961 adult patients between 2003 and 2022. The race of the patients was East Asian (n = 71), Caucasian (n = 71), or Middle Eastern (n = 71). The expected lengths of the NSF and anterior skull base defect were measured and analyzed according to race, age, and sex.

### Results

Compared with Caucasians and Middle Easterners, East Asians had a shorter NSF length ($p < 0.001$) and lower ratio of the expected NSF length to the expected defect length ($p < 0.001$). There was no difference in the values among age groups. The expected NSF length was longer, and the ratio of the expected NSF length to the expected defect length was higher, in males than females ($p < 0.001$ for both).

### Conclusions

East Asians and females had a shorter NSF length and lower ratio of expected NSF to surgical defect lengths after anterior skull base reconstruction compared with the other races and with males, respectively. Anatomical differences should be considered when long NSF lengths are required, such as for anterior skull base reconstruction.

**Data Availability Statement:** All data in this study are openly available on Figshare (https://doi.org/10.6084/m9.figshare.21785837.v1).

**Funding:** This research was supported by the National Research Foundation of Korea (NRF) grant funded by the Ministry of Science and ICT (2021M3F7A1083232). The funders had no role in study design, data collection and analysis, decision to publish, or preparation of the manuscript.

**Competing interests:** The authors have declared that no competing interests exist.

## Introduction

The endoscopic endonasal approach is increasingly being used to treat skull base lesions as a result of improvements in the anatomical knowledge, instruments, navigation techniques, and vascular flaps used for skull base reconstruction [1, 2].

The nasoseptal flap (NSF), introduced by Hadad and Bassagasteguy, is the most commonly used vascularized flap [3]. The flap uses the posterior septal branch of the sphenopalatine artery as a pedicle, and its use reduces the rate of cerebrospinal fluid leakage to < 5% after endoscopic skull base surgery [4]. The NSF is usually harvested from the septal mucosa to the junction of the nasal floor and septum. The NSF surface can be widened by extending the lower incision to the base of the nose or lower nasal passages [5, 6]. However, this method does not extend the NSF length. Although the NSF length is usually sufficient to cover the central skull base defects, such as those after surgery for pituitary lesions, the length may not be sufficient for anterior skull base or posterior fossa defects [7, 8]. Therefore, several methods to extend the NSF length have been developed [9, 10]. If the use of a NSF is expected during surgery, it may be useful to prepare the flap design accordingly. In addition, it may be useful to make an incision in the caudal part of the septal mucosa when harvesting NSFs.

Forensic anthropological studies show that cranial measurements can be used to classify humans by race and geographic origin [11, 12]. Therefore, we hypothesized that the NSF length may be affected by the race, age, and sex of patients. Based on previous studies that performed paranasal sinus computed tomography (PNS CT) to measure the NSF length [7], we performed PNS CT to test our hypothesis. In addition, we investigated whether the conventional NSF length is sufficient to cover an anterior skull base defect, which generally requires the longest NSF length in endoscopic skull base surgery.

## Materials and methods

This retrospective analysis of PNS CT images was approved by the Institutional Review Board of Seoul St. Mary's Hospital (approval no. KC20OISI0461). Between March 2003 and April 2022, 19,961 adult patients underwent PNS CT at our institution. In children and adolescents, the NSF length may change with skull growth. Therefore, only adults were enrolled in this study. A total of 19,650 East Asians, 285 Caucasians, 71 Middle Easterners, 9 South Asians, 4 Hispanics, and 2 Africans were recruited; however, the South Asians, Hispanics, and Africans were excluded from the study because of their small numbers. Because our hospital is located in East Asia, the majority of patients were East Asians; therefore, propensity score matching was performed to allow comparison among races. Propensity score matching was performed between Middle Easterners and the remaining race. We included 71 Middle Easterners, 71 Caucasians, and 71 East Asians. The total 213 patients were enrolled in the study (Fig 1).

We used anatomical landmarks to measure the lengths of the NSF, skull base, and anterior wall of the sphenoid sinus to determine whether the NSF can adequately cover the surgical defect after anterior skull base surgery, similar to a previous study [7]. Table 1 summarizes the definitions, abbreviations, and related illustrations of the indicators measured in this study.

CT studies require identification of anatomical landmarks to allow accurate measurements. In the present study, we selected the projected part of the sphenopalatine foramen (SPF) as the anatomical landmark [7]. To estimate the NSF length required to cover the surgical defect after anterior skull base surgery, the skull base was measured from the posterior wall of the frontal sinus to the upper margin of the anterior wall of the sphenoid sinus (SKB). In addition, the distance from the upper margin of the anterior wall of the sphenoid sinus to the projection of the SPF was measured (SS) (Fig 2). The NSF length required to cover the surgical defect was calculated by adding the measured lengths of SKB and SS.

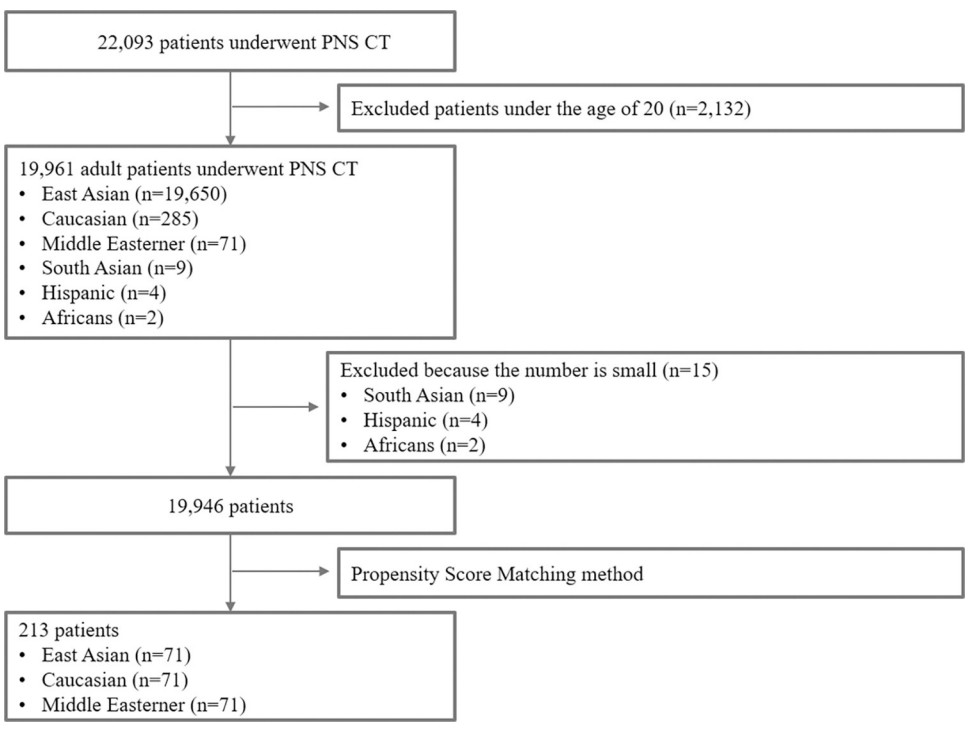

**Fig 1. Flowchart of the study design.**

The NSF length was measured from the most anterior border of the flap to the projection of the SPF (NSF-SPF). We compared this length with the distance from the most anterior to most posterior flap border (NSF-PB) (Fig 3). Then, we calculated the ratio of the length to be covered by the NSF to the expected NSF length.

The CT measurements of the 213 patients were categorized according to race, age, and sex. The patients were divided into the following age groups: 20–40, 41–60, and > 60 years. The data are presented as means ± standard deviation. ANOVA and Tukey's post hoc test were performed to identify significant differences in PNS CT measurements among the groups. Statistical analyses were performed using R software (R Foundation for Statistical Computing, Vienna, Austria). The lengths were measured on PNS CT images using Marosis M-view software (version 5.4; Marotech, Seoul, Korea) and a picture archiving communication system.

**Table 1. Definitions of measurements and their abbreviations, related illustrations.**

| Measurements | Abbreviation | Figure |
|---|---|---|
| From the posterior wall of the frontal sinus to the upper margin of the anterior wall of the sphenoid sinus | SKB | Fig 2(A) |
| From the upper margin of the anterior wall of the sphenoid sinus to the SPF projection | SS | Fig 2(B) |
| SPF to posterior wall of frontal sinus through anterior wall of sphenoid sinus | SKB+SS | Fig 2(A)+2(B) |
| From the most anterior margin of the NSF to the SPF projection | NSF-SPF | Fig 3(A) |
| From the most anterior margin to the posterior margin of the NSF | NSF-PB | Fig 3(B) |

SKB, skull base; SS, sphenoid sinus; NSF, nasoseptal flap; SPF, sphenopalatine foramen; PB, posterior border.

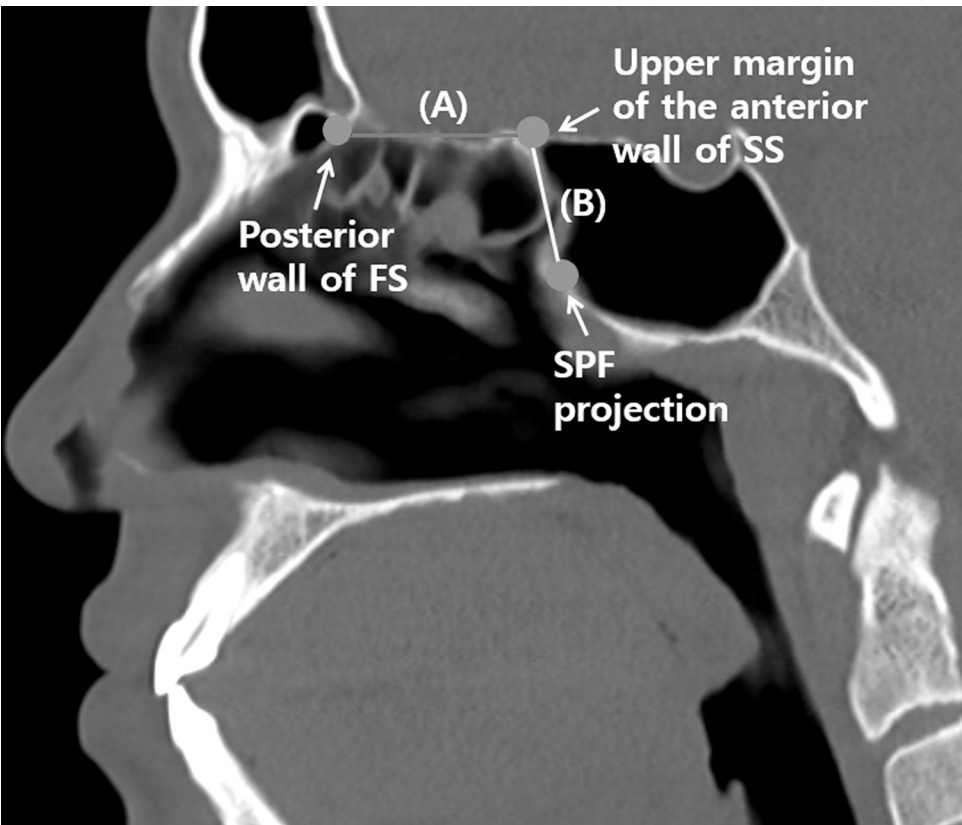

**Fig 2. Measurement of the lengths of the skull base and anterior wall of the sphenoid sinus in the sagittal plane on PNS CT images.** Distance from the posterior wall of the frontal sinus to the upper margin of the anterior wall of the sphenoid sinus (A). Distance from the upper margin of the anterior wall of the sphenoid sinus to the SPF projection (B). FS, frontal sinus; SB, skull base; SS, sphenoid sinus; SPF, sphenopalatine foramen.

## Results

The mean ages of the East Asian, Caucasian, and Middle Eastern patients were 42.1 (range, 20–73), 44.3 (range, 21–67), and 41.6 (range, 20–76) years, respectively. Of the 71 East Asian, Caucasian and Middle Eastern patients each, there were 34 males (47.9%) and 37 females (52.1%). There was no significant difference among the races in terms of age or sex. In the SKB, there were shorter East Asian patients than Middle Eastern patients ($p = 0.008$). There was no significant difference between the races in lengths of SS ($p = 0.277$) or SKB+SS ($p = 0.052$). By contrast, the NSF-SPF ($p < 0.001$), NSF-PB ($p < 0.001$), NSF-SPF/SKB+SS ($p = 0.001$), and NSF-PB/SKB+SS ($p < 0.001$) of East Asian were significantly smaller than the other two races. However, these factors did not differ significantly between Caucasians and Middle Easterners (Table 2).

Table 3 shows a comparison of the variables among the three age groups. Of the patients aged 20–40 years (n = 86), there were 41 (47.7%) males and 45 (52.3%) females and 31 (36.0%), 25 (29.1%), and 30 (34.9%) East Asians, Caucasians, and Middle Easterners, respectively. Of the patients aged 41–60 years (n = 80), there were 34 (42.5%) males and 46 (57.5%) females and 24 (30.0%), 31 (38.8%), and 25 (31.2%) East Asians, Caucasians, and Middle Easterners, respectively. Of the patients aged > 60 years (n = 47), there were 27 (57.4%) males and 20 (42.6%) females and 16 (34.0%), 15 (32.0%), and 16 (34.0%) East Asians, Caucasians, and Middle Easterners, respectively. There was no significant difference in sex or race among the

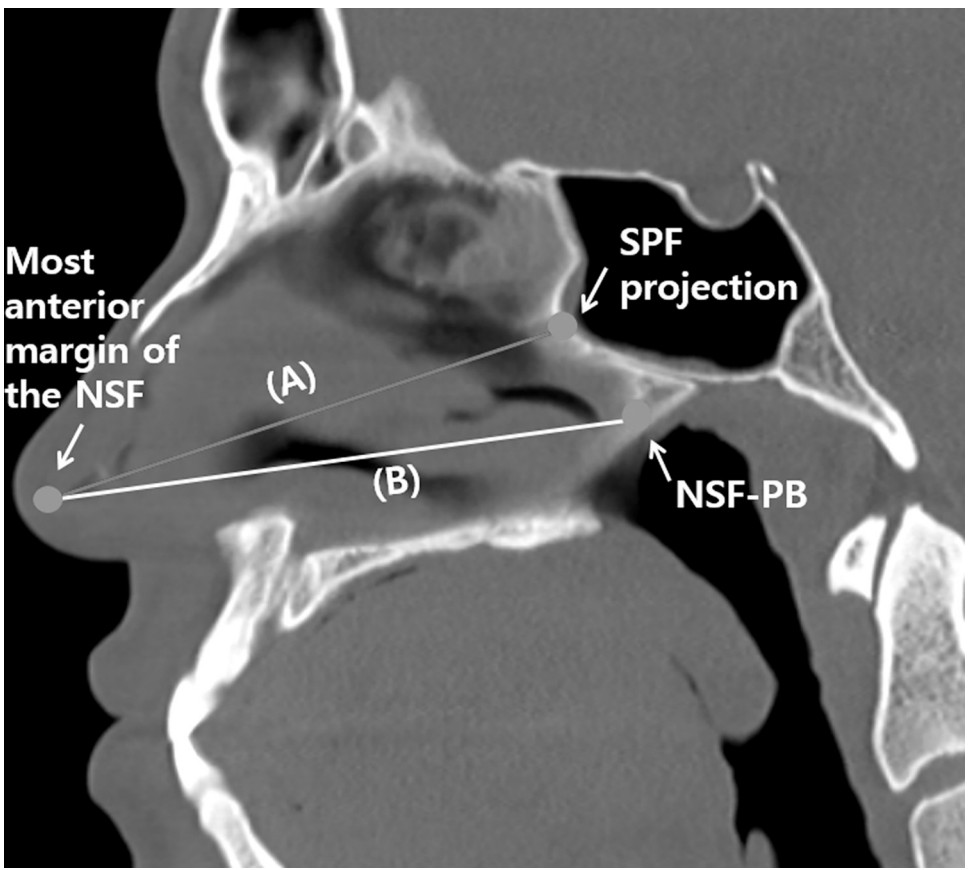

**Fig 3. Measurement of the NSF length in the sagittal plane on PNS CT images.** Distance from the most anterior margin of the NSF to the SPF projection (A). Distance from the most anterior to the posterior margin of the NSF (B). NSF, nasoseptal flap; SPF, sphenopalatine foramen; PB, posterior border.

**Table 2. Summary of patient characteristics and CT measurements by race.**

|  | East Asian (n = 71) | Caucasian (n = 71) | Middle Easterner (n = 71) | *p*-value |
|---|---|---|---|---|
|  | Mean±SD | Mean±SD | Mean±SD |  |
| Age (yr) | 41.77±15.80 | 44.37±13.46 | 41.61±16.14 | 0.478 |
| Gender |  |  |  |  |
| Male | 34 | 34 | 34 |  |
| Female | 37 | 37 | 37 |  |
| SKB | 28.00±4.09 | 29.93±4.48 | 29.91±3.96 | 0.008* |
| SS | 17.10±2.84 | 16.50±2.28 | 17.05±2.27 | 0.277 |
| SKB+SS | 45.10±4.72 | 46.42±4.96 | 46.96±4.28 | 0.052 |
| NSF-SPF | 68.16±5.73 | 74.76±5.34 | 75.24±6.17 | <0.001* |
| NSF-PB | 74.41±5.77 | 82.36±5.13 | 83.05±6.63 | <0.001* |
| NSF-SPF / SKB+SS | 1.52±0.15 | 1.63±0.19 | 1.61±0.17 | 0.001* |
| NSF-PB / SKB+SS | 1.66±0.15 | 1.79±0.21 | 1.78±0.20 | <0.001* |

* *p* <0.05 for the test, SD; standard deviation.

**Table 3. Summary of patient characteristics and CT measurements by age.**

| | 20–40 years old (n = 87) | 41–60 years old (n = 79) | Over 61 years old (n = 47) | p-value |
|---|---|---|---|---|
| | Mean±SD | Mean±SD | Mean±SD | |
| Gender | | | | 0.292 |
| Male | 41 | 34 | 27 | |
| Female | 46 | 45 | 20 | |
| Race | | | | 0.931 |
| East Asian | 32 | 23 | 16 | |
| Caucasian | 25 | 31 | 15 | |
| Middle Easterner | 30 | 25 | 16 | |
| SKB | 29.11±4.27 | 28.75±4.28 | 30.48±4.14 | 0.078 |
| SS | 16.90±2.64 | 17.01±2.36 | 16.63±2.39 | 0.716 |
| SKB+SS | 46.01±4.75 | 45.76±4.26 | 47.12±5.28 | 0.274 |
| NSF-SPF | 71.70±6.43 | 72.50±6.44 | 74.95±6.72 | 0.022* |
| NSF-PB | 78.97±6.91 | 79.95±6.95 | 81.70±7.26 | 0.101 |
| NSF-SPF / SKB+SS | 1.57±0.17 | 1.60±0.19 | 1.60±0.17 | 0.495 |
| NSF-PB / SKB+SS | 1.73±0.19 | 1.76±0.21 | 1.75±0.19 | 0.605 |

SD; standard deviation.

age groups. In addition, SKB, SS, SKB+SS, NSF-PB, NSF-SPF/SKB+SS, and NSF-PB/SKB+SS were not significantly different among the groups.

The 102 males had a mean age of 44.3 (range, 21–67) years and comprised 34 (33.3%) East Asians, 34 (33.3%) Caucasians, and 34 (33.3%) Middle Easterners. The 111 females had a mean age of 42.0 (range, 20–76) years and comprised 37 (33.3%) East Asians, 37 (33.3%) Caucasians, and 37 (33.3%) Middle Easterners. There was no significant difference in age or race between males and females. The SS length was shorter in the female than male patients ($p = 0.009$). However, there was no significant difference in SKB ($p = 0.822$) or SKB+SS ($p = 0.091$) between the sexes. By contrast, males had larger values of NSF-SPF, NSF-PB, NSF-SPF/SKB+SS, and NSF-PB/SKB+SS compared with females ($p < 0.001$) (Table 4).

**Table 4. Summary of patient characteristics and CT measurements by sex.**

| | Male (n = 102) | Female (n = 111) | p-value |
|---|---|---|---|
| | Mean±SD | Mean±SD | |
| Age (yr) | 42.74±15.46 | 42.43±14.96 | 0.885 |
| Race | | | |
| East Asian | 34 | 37 | |
| Caucasian | 34 | 37 | |
| Middle Easterner | 34 | 37 | |
| SKB | 29.42±4.51 | 29.15±4.03 | 0.635 |
| SS | 17.37±2.59 | 16.43±2.30 | 0.006* |
| SKB+SS | 46.80±4.75 | 45.58±4.61 | 0.059 |
| NSF-SPF | 76.77±5.59 | 68.98±5.06 | <0.001* |
| NSF-PB | 83.97±5.83 | 76.22±5.94 | <0.001* |
| NSF-SPF / SKB+SS | 1.66±0.19 | 1.52±0.14 | <0.001* |
| NSF-PB / SKB+SS | 1.81±0.20 | 1.68±0.17 | <0.001* |

* $p < 0.05$ for the test, SD; standard deviation.

## Discussion

An increasing number of cranial base surgeries are being performed using the endoscopic endonasal approach. The endoscopic endonasal transsphenoidal approach is preferred for pituitary gland surgery [13, 14]. Furthermore, endoscopic endonasal skull base surgery via the transcribriform corridor is increasingly being performed for the treatment of anterior skull base lesions, such as olfactory groove meningioma and esthesioneuroblastoma [15–22]. Prior to the development of endoscopic surgery, frontal craniotomy was performed to allow good visualization of the anterior cranial fossa. The operation was performed using a transcranial approach, and the defect was reconstructed using a pericranial flap [23]. Expanded endoscopic endonasal procedures are increasingly being performed for the treatment of lesions beyond the sella turcica because of the reduced operating time and complications compared with surgery using the open approach, such as craniotomy [20, 24–26]. The NSF length is usually adequate for bony defects created during endoscopic endonasal transsphenoidal surgery. Therefore, several methods were developed to expand the NSF width, which did not increase the NSF length [5, 6]. However, during endoscopic endonasal skull base surgery via the transcribriform corridor, reconstruction of the anterior skull base defects may be difficult due to the short NSF length [17]. As a result, 17–30% of patients experience CSF leakage after endoscopic transcribriform surgery [17–19, 27].

Previous studies used CT to measure the NSF length and area that can be harvested and evaluated whether the length and area are sufficient to cover the potential skull base defect [7]. The CT measurements showed that the NSF covered the defects created by anterior skull base, transsellar/transplanar, or transclival endoscopic endonasal surgery, but not combined defects. However, during surgery, the NSF cannot be applied in a completely straight manner to avoid excessive tension on the flap during repositioning, which leads to a shorter NSF length available than that measured by CT. To cover the defect created during anterior skull base surgery, the NSF is raised as far forward as possible to reach the posterior table of the frontal sinus (anterior boundary of the skull base defect). This process creates tension in the NSF, which prevents the flap from reaching sufficiently forward and causes it to retreat in the opposite direction from the posterior table of the frontal sinus. Additionally, excessive stretching of the pedicled NSF prevents adequate contact of the flap to the posterior side of the skull base defect [9]. However, the use of excessively large NSFs may lead to additional complications, such as flap necrosis [28]. Harvesting a greater flap area from the nostril region leads to longer time required for re-epithelialization of the exposed cartilage and crust formation on the bone, which may damage the internal nasal valve and cause nasal congestion [29]. Harvesting a greater flap area from the cribriform plate may cause olfactory dysfunction and nasal deformities, such as dorsal nasal collapse and septal perforation [30–32]. Several procedures have been developed to prevent these complications and extend the NSF length. In one such procedure, a slit incision is made on the sphenoidal segment of the NSF to eliminate the tension in the NSF and allow it to cover the anterior skull base defect [9]. Another procedure improves NSF mobility by dissecting and extending the NSF pedicle to the inside of the pterygopalatine fossa [10]. However, this procedure can also cause complications, such as flap necrosis, subdural empyema in the resection bed, and bleeding due to excessive dissection around the pedicle. In cases of an insufficient NSF length, the defect cannot be completely covered, whereas an excessive NSF length is associated with a higher risk of complications. Therefore, it is necessary to use an appropriate flap size according to the individual characteristics of each patient.

Forensic anthropologists have taken the biological race concept of classical physical anthropology and applied it to human identification methods [12]. In forensic anthropology, why and how the concept of biological race works was considered by Sauer [33], who hypothesized

that American forensic anthropologists do a good job at what they do because the social races and skeletal morphologies of American whites and blacks match. Forensic anthropology studies show that cranial measurements can be used to accurately classify humans by geographic origin [34]. Based on this, the role of forensic dentistry and anthropology has been emphasized in the identification of human remains [11], and in the medical field, many attempts have been made to reveal differences between races through various measurement methods including lateral cephalogram. There was a study comparing the skull morphology of Asians and Caucasians in crouzon syndrome [35], and a study comparing craniofacial differences in skeletal malocclusion between Japanese and British Caucasians [36]. Our study also investigated the anatomical differences according to race, gender, and age.

Previous studies that used CT [7] or cadavers [37] to measure the skull base parameters did not consider the effects of race, age, or sex. Therefore, we classified the parameters measured in this study according to race, age, and sex. We measured the parameters on preoperative PNS CT images because these data are the most commonly available.

SKB was significantly smaller in East Asians than Caucasians and Middle Easterners. However, there was no difference among the races in terms of SKB+SS, which indicates the length of the defect. Both NSF-SPF and NSF-PB were smaller in East Asians than other races. The NSF-SPF/SKB+SS and NSF-PB/SKB+SS values, which are the ratios of the defect length to the expected flap length, were significantly smaller in East Asians than other races. Our results showed that Middle Easterners are more anatomically similar to Caucasians than East Asians. The racial anatomical differences investigated through these studies can be helpful to the field of forensic anthropology.

Patients aged over 20 years did not exhibit any significant differences in the CT parameters according to age. Although skull growth is completed by the age of 20 years, after which the skull undergoes degeneration over time, the CT parameters did not change with age.

The SS values was smaller in females than males. However, SKB+SS was not significantly different between the sexes. By contrast, the NSF-SPF, NSF-PB, NSF-SPF/SKB+SS and NSF-PB/SKB+SS values were significantly lower in females than males. Therefore, there was no significant difference in the length of the skull base between males and females. By contrast, the NSF length that can be harvested is shorter in females than males.

Our results suggest that NSFs should be harvested more aggressively during anterior skull base surgery in East Asians and females. Regardless of race, age, or sex, NSF-SPF/SKB+SS and NSF-PF/SKB+SS can be measured on preoperative CT images and compared with the mean values reported in the present study; if lower than the mean value, the anterior and posterior margins of the harvested NSF should be extended. Additionally, a sphenoidal slit incision or pedicle dissection of the internal maxillary artery can be performed [9, 10].

Our study had some limitations. First, several parameters were measured using the straight length on PNS CT images, which may differ from the actual flap. The flap is mobile, and its length differs depending on the tension. Second, the mucosal thickness and degree of flap rotation may vary among individuals. However, CT is the most commonly performed test prior to surgery, and the parameters are difficult to evaluate by other investigations; therefore, we measured the parameters by CT. Third, a landmark was selected posterior to the frontal sinus recess, which may lead to shorter measurements compared with the actual defect. A radioanatomical study showed that the use of a prominent landmark improves the objectivity of measurements. We selected the posterior part of the frontal sinus recess as a suitable landmark in accordance with a previous study [7]. Fourth, if the surgical defect extends from the anterior skull base to the upper skull base of the sphenoid sinus due to a large meningioma-like tumor, the anterior wall of the sphenoid sinus may be opened. In such cases, multiple NSFs may be required to cover the anterior skull base defect and posterior wall of the sphenoid sinus.

However, we did not measure this length, which should be measured in future studies. Fifth, we matched the controls and cases by age and sex, despite the risk of selection bias. Sixth, children were excluded from the analysis because the NSF length may change with skull growth. Finally, only East Asian, Caucasian, and Middle Eastern races were analyzed because of the small number of patients of other races. Future studies should include patients of various age groups (including children) and races.

## Conclusions

The expected NSF length is shorter in East Asians than Caucasians and Middle Easterners. Additionally, the NSF length is shorter in females than males. Therefore, when long NSF length are required, such as for anterior skull base reconstruction, the individual anatomical differences should be considered.

## Acknowledgments

We wish to thank Dr. Kim for his help with data processing and all participants for supporting our research.

## Author Contributions

**Conceptualization:** Jin Hee Cho, Sung Won Kim.

**Data curation:** Chang Yeong Jeong, Sung Won Kim, Jae-Sung Park, Mohammed Abdullah Basurrah.

**Formal analysis:** Jin Hee Cho, Jae-Sung Park, Mohammed Abdullah Basurrah.

**Funding acquisition:** Do Hyun Kim.

**Investigation:** Jin Hee Cho, Yong Jin Park, Mohammed Abdullah Basurrah.

**Project administration:** Sung Won Kim, Mohammed Abdullah Basurrah, Soo Whan Kim.

**Resources:** Do Hyun Kim.

**Software:** Jae-Sung Park, Do Hyun Kim.

**Supervision:** Jin Hee Cho, Yong Jin Park, Soo Whan Kim.

**Writing – original draft:** Chang Yeong Jeong.

**Writing – review & editing:** Do Hyun Kim, Soo Whan Kim.

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
