## [Decision Letter · Decision Letter 0]

23 Nov 2022

PONE-D-22-26727Differences in the predicted nasoseptal flap length among races: A propensity score matching analysisPLOS ONE

Dear Dr. Kim,

Thank you for submitting your manuscript to PLOS ONE. After careful consideration, we feel that it has merit but does not fully meet PLOS ONE’s publication criteria as it currently stands. Therefore, we invite you to submit a revised version of the manuscript that addresses the points raised during the review process.

We look forward to receiving your revised manuscript.

Kind regards,

Johari Yap Abdullah, B.S. & I.T, GradDip ICT, M.Sc, Ph.D.

Academic Editor

PLOS ONE

Journal Requirements:

“This research was supported by the National Research Foundation of Korea (NRF) grant funded by the Ministry of Science and ICT (2021M3F7A1083232).”

“This research was supported by the National Research Foundation of Korea (NRF) grant funded by the Ministry of Science and ICT (2021M3F7A1083232). The sponsors had no role in the study design, data collection and analysis, decision to publish, or preparation of the manuscript.”

“This research was supported by the National Research Foundation of Korea (NRF) grant funded by the Ministry of Science and ICT (2021M3F7A1083232).”

Reviewers' comments:

Reviewer's Responses to Questions

**Comments to the Author**

1. Is the manuscript technically sound, and do the data support the conclusions?

Reviewer #1: Yes

Reviewer #2: Yes

2. Has the statistical analysis been performed appropriately and rigorously? 

Reviewer #1: Yes

Reviewer #2: Yes

3. Have the authors made all data underlying the findings in their manuscript fully available?

Reviewer #1: Yes

Reviewer #2: No

4. Is the manuscript presented in an intelligible fashion and written in standard English?

Reviewer #1: Yes

Reviewer #2: Yes

5. Review Comments to the Author

Reviewer #1: This article analyzes the difference of nasoseptal flep length between races and genders which is important during endoscopic sinus surgery. The figures and tables of the article are accurate and the language & article design is understandable.

Reviewer #2: Dear Author,

Thank you for the wonderful work on the NSF. I did enjoy reading it. There are some issues which needs to be addressed, before this article is publication ready. Issues are addressed below:

1. While the authors mention full data availability is provided. No attachment/link is provided as part of the manuscript. Hence it is unknown if full data availability is granted. Please read the terms and conditions regarding PLOS data policy for more information.

2. There is systematic error between the word and the reference ([]). There should be a space between the word and the bracket and not immediately bracket. Please fix for all. Please refer to other successful articles published as example of correct citation method.

3. Need to add comparison between other population data research for same or similar region in the introduction and discussion. The main aim of the paper is to compare between lengths of the of the NSF between race, age and skull. The paper is more related to Forensic Anthropology population data as any measurement involving nasal projection will lead to difference in population. Caucasoid, Mongoloid and Negroid type skull. The reason why you obtain similar values for Caucasians and Middle Eastern is because, they come from the same skull class. Please read and add more to the introduction.

4. Make abbreviations available for Table 1 and Figure 2 and 3. For Table 1, PB, SS, SKB abbreviations need to be made available below the text. For Figure 2 and 3, please state all the points used for measurement. Example in Figure 2, the intersection between (A) and (B) is not not mentioned, making the Figure not self-explanatory. Same with Figure 3 at the tip of the nose.

5. Methods and materials. While the reason to exclude one data point (East Asian=70) is reasonable when there are limited samples. For this research I would urge maintaining it to 71 because of the 19,650 data of East Asians. There is enough data to choose and run the experiment again. If it were from the Middle Eastern group, then it would be acceptable. I would also recommend deleting the sentence line 74-76 after adding a new data point. Retaining the sentence as it is, does not reflect good diligence.

6. Discussion. Similar to point 3, need to add some paragraphs relating to differences in Caucasian, Mongoloid and Negroid-type skull. The reason why there are little references in the discussion is also related to this. The authors need to explore more literature review regarding skull morphology and race. There are some papers which make use of lateral cephalogram articles as well.

Based on the comments, I would invite the author to revamp the manuscript first before making a submission again. Hence, it will be better to reject the article for now to give more time to the authors to revamp the manuscript.

6. PLOS authors have the option to publish the peer review history of their article (what does this mean?). If published, this will include your full peer review and any attached files.

Reviewer #1: **Yes: **Demet Yazici

Reviewer #2: No

---

## [Author Response · Author response to Decision Letter 0]

3 Jan 2023

The reviewer's accurate point was an opportunity to review the article once more. I wrote down a specific answer through " response to reviewers". Thank you.

---

## [Decision Letter · Decision Letter 1]

3 Mar 2023

Differences in the predicted nasoseptal flap length among races: A propensity score matching analysis

PONE-D-22-26727R1

Dear Dr. Kim,

We’re pleased to inform you that your manuscript has been judged scientifically suitable for publication and will be formally accepted for publication once it meets all outstanding technical requirements.

Kind regards,

Johari Yap Abdullah, B.S. & I.T, GradDip ICT, M.Sc, Ph.D.

Academic Editor

PLOS ONE

Additional Editor Comments (optional):

Reviewers' comments:

Reviewer's Responses to Questions

**Comments to the Author**

1. If the authors have adequately addressed your comments raised in a previous round of review and you feel that this manuscript is now acceptable for publication, you may indicate that here to bypass the “Comments to the Author” section, enter your conflict of interest statement in the “Confidential to Editor” section, and submit your "Accept" recommendation.

Reviewer #2: All comments have been addressed

2. Is the manuscript technically sound, and do the data support the conclusions?

Reviewer #2: Yes

3. Has the statistical analysis been performed appropriately and rigorously? 

Reviewer #2: Yes

4. Have the authors made all data underlying the findings in their manuscript fully available?

Reviewer #2: Yes

5. Is the manuscript presented in an intelligible fashion and written in standard English?

Reviewer #2: Yes

6. Review Comments to the Author

Reviewer #2: The authors have addressed all the problems of the original article, and the article now can be published.

7. PLOS authors have the option to publish the peer review history of their article (what does this mean?). If published, this will include your full peer review and any attached files.

Reviewer #2: No

---

## [Editor Report · Acceptance letter]

7 Mar 2023

PONE-D-22-26727R1 

Differences in the predicted nasoseptal flap length among races: A propensity score matching analysis 

Dear Dr. Kim:

I'm pleased to inform you that your manuscript has been deemed suitable for publication in PLOS ONE. Congratulations! Your manuscript is now with our production department. 

Kind regards, 

on behalf of

Dr. Johari Yap Abdullah 

Academic Editor

PLOS ONE